# Crime Analysis of the Metropolitan Region of Santiago de Chile: A Spatial Panel Data Approach

Pablo Cadena-Urzúa [1], Álvaro Briz-Redón [2] and Francisco Montes [2,*]

1    Doctoral Programme in Law, Political Science and Criminology, Department de Dret Penal, Universitat de València, 46022-València, Spain
2    Department d'Estadística i I O., Universitat de València, 46100-Burjassot, Spain
*    Correspondence: francisco.montes@uv.es

**Abstract:** The aim of our work is to determine the influence that socio-economic and demographic factors have had on crimes that have taken place during the period 2010–2018 in the communes of the Metropolitan Region of Chile, as well as the existence of possible spatial or temporal effects. We address 12 kinds of crime that we have grouped into two main types: against people and against property. Our interest focuses on crimes against people, using crimes against property as an additional covariate in order to investigate the existence of the broken-windows phenomenon in this context. The model chosen for our analysis is a spatial panel model with fixed effects. The results highlight that covariates such as infant mortality, birth rate, poverty and green areas have a significant influence on crimes against people. Regarding the spatio-temporal covariates, one effect observed is that there is a displacement of crime towards neighbouring communes, leaving open a new line of study to discover the causes of this displacement.

**Keywords:** crime analysis; crime prevention; evidence-based crime prevention; spatial panel





## 1. Introduction

Crime and violence in the Latin America and Caribbean region are pervasive and costly, especially violent crimes (Chioda 2016). It can be asserted that, in Latin America, there is a historical relationship between the city, its urbanisation rates and violence (García and Ramos 2019; Gaviria and Pagés 1999; Velásquez and García Pinzón 2008). Similarly, as Carrión (2008) asserts, Latin America has become a continent of cities; accelerated urbanisation has been accompanied by rising crime rates. Analogously, people interviewed by Corporación Latinobarómetro for its 2015 report (Corporación Latinobarómetro 2016) cited crime as the second most worrying issue, with 25% of mentions, economic issues being the first at 37%. Recently, in their survey for 2020, they pointed out that street violence was the most frequent. The countries where respondents perceive the most street violence are Argentina (58%), Uruguay and Chile (56%). Crime has therefore become a major problem and a priority for governments. This type of crime causes great social alarm in developed countries (Cole and Gramajo 2009), consequently, government authorities consider its prevention among their most important objectives, which requires an understanding of the factors that determine crime in order to contain it.

The motivation to address crime from the southern country of Chile stems from the fact that this nation has been a benchmark in various aspects in the region. For example, the International Monetary Fund (IMF 2022) projects that Chile will reach a GDP per capita of US$30,000 within two years, making it the first country in South America to reach figures similar to those of several European nations, as well as bringing poverty rates down to some of the lowest levels on the continent. On the crime front, according to the report of InSight Crime Foundation (2022), Chile is the safest country in Latin America (measured by homicide rate), as a result of having implemented several important reforms over the

last two decades, which have allowed for the establishment of efficient security forces, an independent judicial system and impactful public policies. Nevertheless, studies such as the one presented in this paper should help to improve the design of public policies by identifying the factors that influence crime.

As a result of the progressive increase in studies in the discipline, there is more evidence to discuss criminal behaviour. The models with the greatest empirical contribution currently postulate multicausality and multifactoriality in the explanation of criminal behaviour (Cardona 2020). These analyses include biological, social, familial and individual variables (Taylor 2016). Since the mid-1990s, several ecological studies on crime have attempted to find the political-institutional and socioeconomic factors that are configured in urban settings in Latin America where there are high rates of violence, comparing the distribution of homicides in different aggregate units: countries, regions, provinces, departments, metropolitan areas, cities, communes and neighbourhoods (Beato et al. 2004; Dammert and Reyes 2004; García 2016; García and Ramos 2019).

As Wortley (1997) states, behavior varies to a greater or lesser extent depending on the context in which it occurs. Several empirical studies have considered the effects of socio-economic and demographic conditions as factors in crime rates, just as Brunton-Smith and Sturgis (2011); Farrell et al. (2000); Krug et al. (2002) and Vieno et al. (2013), who highlight the negative socioeconomic and community attributes of neighborhoods as precipitants of crime, in addition to that which was sustained by Fajnzylber et al. (2000), whose work includes a few countries in the region, but takes into account fewer potential determinants than the present research. Additionally, the relationship between poverty and crime has been studied in Sharkey et al. (2016), Anser et al. (2020) and Dong et al. (2020). Additionally, the influence of infant mortality, an indirect measure of poverty, on crime has been analysed in Messner et al. (2010).

According to Núñez et al. (2003), adequate knowledge of the determinants of crime at regional level can be of great value in diagnosing the crime situation. Later on, de la Fuente and Mejías-Navarro (2011) argue that explanatory variables such as inequality, drug use, poverty, schooling rate, as well as the type of leadership prevailing in the household influence violent property crime behaviour. These two works are the closest to the present study in that they analyse crime in Chile. However, they differ from our study in that they include fewer factors, variables and possible determinants, as well as employ a different statistical technique. Employing more determinants or covariates in the model allows us to improve the analysis in several ways. On the one hand, this can help to increase the overall goodness of fit of the model and to find the determinants involved in crime rates, with the consequent practical interest. On the other hand, the fact that a greater proportion of the variability in the data is explained by environmental or socioeconomic determinants allows us to obtain more realistic estimates of the impact of the covariates of greatest interest to the study, which in our case are those related to the broken-windows phenomenon. The absence of a comprehensive set of determinants unrelated to these variables could lead to an overestimation of the importance of this phenomenon.

Taking the aforementioned documents as a premise, our article aims to determine the influence that these factors, both socioeconomic and demographic, have on crime, specifically those crimes that cause the greatest social alarm (Cole and Gramajo 2009; Fiscalía de Chile 2021) in developed countries. This research covers the 52 communes (municipalities) that make up the Metropolitan Region of Santiago de Chile (MRS), over almost a decade of background; as additional dimensions, we consider the spatial and temporal components of the data, which are the elements which differentiate this from previous works and provide great value in crime theory. As Ye and Wu (2011) point out, incorporating time into spatial patterns allows for better modelling of phenomena than cross-sectional data configuration (Elhorst 2003, 2009).

In a recent article, Bivand et al. (2021) review the spatial econometrics packages available for the R software (R Core Team 2020). They illustrate the different methods with real-world examples, including crime data for the 90 counties of North Carolina over the

period 1981-1987. Among the libraries described in the article, there is splm library, which allows the fitting of different models to spatial panels. Details of the library can be found in Millo and Piras (2012). Our interest in this library is because spatial panels are a suitable tool for the processing of MRS data. Thus, Ye and Wu (2011) analyses the dynamics of homicide rate patterns in Chicago in the period 1960-1995 using spatial panel, Lauridsen et al. (2013) investigate the factors affecting crime rates in the EU-15 countries over the years 2000 to 2007, with special attention to the inflation rate, education levels, income, and employment. In contrast to the previous ones, which work with per capita crime ratios and thus avoid population, Liesenfeld et al. (2017) model monthly major crime counts at the census tract level in Pittsburgh, using a spatio-temporal Poisson panel model whose ML estimates are obtained. In the same direction, Glaser et al. (2021) propose a new spatial panel regression framework with fixed effects to provide consistent count predictions, taking into account their integer and non-negative nature. They test their model with the same monthly crime data for Pittsburgh used by Liesenfeld et al. (2017).

The paper is organized as follows: In Section 2, we present the MRS crime data with a descriptive analysis of these. Section 3 describes the spatial panel models, with special attention to the one that will be applied to our data. The results obtained with this model are shown in Section 4. Two sections devoted to Discussion and Bibliography close the article.

## 2. Data

The data that we are going to analyse, as mentioned in the Introduction, refer to the crimes that cause great social alarm and that have taken place in the 52 communes of the MRS during the period 2010–2018. The 12 kinds of crime are shown in Table 1. The data were obtained from the Instituto Nacional de Estadística (INE) de Chile and the Subsecretaría de Prevención del Delito and Sistema Nacional de Información Municipal (SINIM).

**Table 1.** Crimes collected in the database.

| | |
|---|---|
| 1 Homicide (murder) | 7 Vehicle theft |
| 2 Robbery | 8 Burglary in an residential location |
| 3 Injuries | 7 Burglary in a non-residential location |
| 4 Other thefts with force | 10 Objects theft of or from vehicles |
| 5 Robbery by intimidation | 11 Robbery by surprise |
| 6 Robbery with violence | 12 Rape |

The database contains 2,142,963 records with 14 variables in each (see Table 2). All the information collected in the database refers to crime and victim, except for the variable arrested, dichotomous, which indicates whether or not the offender was arrested.

**Table 2.** Variables in the database.

| | |
|---|---|
| commune | day of the month |
| quadrant | day of the year |
| location | time of day |
| crime | arrested |
| year | sex |
| month | education |
| week | age |

This information is completed with a file containing yearly data from the communes. This consists of 468 records (52 communes for 9 years), each one with 16 variables (Table 3).

Crime data are subjected to a first aggregation process by accumulating them by commune and year. These aggregated data are matched with the commune database. A second aggregation process takes place with the crimes (Table 4), which are in turn grouped into crimes *against people* and crimes *against property*. This file constitutes the work database, a balanced spatial panel data.

**Table 3.** Covariables containing information about the communes.

| Variable | Description |
| --- | --- |
| year | year |
| commune | name of the commune |
| area | communal area in km$^2$ |
| density | population density |
| population | communal population estimated by INE |
| women | percentage of female communal population |
| infant.mort | infant mortality rate |
| birth.rate | birth rate |
| social.prog | share of social programme area in total budget (%) |
| cult.prog | share of cultural programmes in total budget (%) |
| ipp | permanent own income per capita (%) |
| school.att | percentage of communal school attendance |
| reb | school failure rate in basic education (communal) |
| rem | school failure rate in secondary education (communal) |
| poverty | percentage of the population living in poverty |
| green.areas | m$^2$ of green areas per capita |

Before beginning with a brief description of the data, it is appropriate to point out the potential sources of error in the crime data records: from the incorrect location of a crime, the incidence of which in our case would be minor as we work with aggregate data, to that derived from unreported crimes. With regard to Chile, in a document published by the Biblioteca del Congreso Nacional de Chile (Biblioteca del Congreso Nacional de Chile 2014), the estimated proportion of non-reporting in 2013 is 56.1%, with significant variations depending on the type of crime, this being much lower for crimes that cause great social alarm.

The underreporting of crime-related events is an issue that can limit or bias the results of our analysis. Quantifying the impact of underreporting on statistical modeling is highly complex. In a recent paper, however, Buil-Gil et al. (2022) have concluded through a simulation study that the risk of bias in the estimation of crime data models is greater over small geographies than over large ones. Hence, the use of crime data on large geographic units is more convenient in this regard. In our case, the 52 municipalities of the MRS can be considered as large geographical units, as they are in fact larger in size (on average) than UK wards, which acted as the largest units in the study of Buil-Gil et al.

**Table 4.** Classification of crimes.

| Crimes against People | Crimes against Property |
| --- | --- |
| homicide (murder) | robbery |
| injuries | other thefts with force |
| robbery by intimidation | vehicle theft |
| robbery with violence | burglary in an residential location |
| robbery by surprise | burglary in a non-residential location |
| rape | object theft of or from vehicles |

This aggregation of crimes distinguishes between the most violent crimes, which have people as victims and cause great social alarm, and those which, although alarming, are less violent. The purpose of this division is to check whether the *broken-windows* phenomenon also occurs in our spatio-temporal context. This theory states that the intensity of less serious crime in a given space is a leading indicator of more serious crimes (Wilson and George 1982). In our case, property crimes will be those considered minor crimes, which would precede violent crimes. Table 5 shows the annual evolution of both types of crimes.

**Table 5.** Distribution of crimes by type (see Table 4) and year.

| Year | Against People | Against Property | Totals |
|---|---|---|---|
| 2010 | 95,444 | 145,238 | 240,682 |
| 2011 | 104,240 | 155,665 | 259,905 |
| 2012 | 89,292 | 142,298 | 231,590 |
| 2013 | 93,903 | 145,912 | 239,815 |
| 2014 | 98,276 | 149,384 | 247,660 |
| 2015 | 98,221 | 146,471 | 244,692 |
| 2016 | 91,985 | 132,557 | 224,542 |
| 2017 | 96,269 | 131,992 | 228,261 |
| 2018 | 101,366 | 124,450 | 225,816 |
| totals | 868,996 | 1,273,967 | 2,142,963 |

In the subsequent analysis instead of crime counts, their ratios per 10,000 inhabitants are used. The database is completed with these ratios and with their spatial lag and the spatio-temporal lag of order (1.1) for both types of crimes, i.e., the average of the ratios of crimes that had occurred in the previous year in the commune neighborhood.

Figure 1 shows the spatial distribution of the average ratio over the years 2010–2018 of crimes against people and property. Although this is an average value for the whole period, the presence of spatial autocorrelation is perceived, and is confirmed by the graph of the Moran indices (Moran 1950) in Figure 2 (left), all of which are significant for both types of crime. The evolution of the annual average of the communal ratios is shown in the right graph on Figure 2. Concerning this figure, it is convenient to note that, in line with the last 100 years in jurisdictions around the world, there is a significant downwards trend in crime rates per 10,000 population.

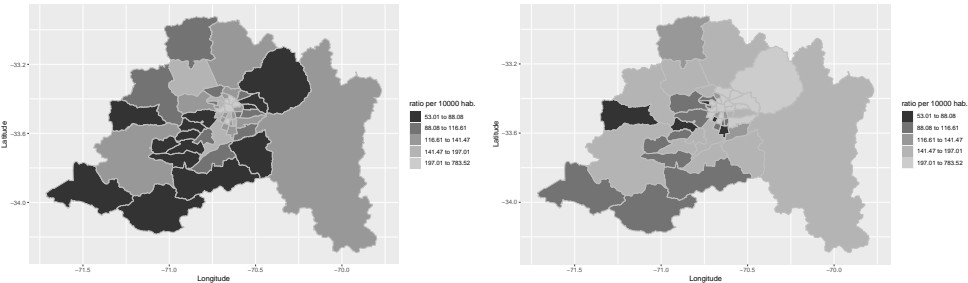

**Figure 1.** Average annual ratio of crimes against people (**left**) and crimes against property (**right**) (2010–2018).

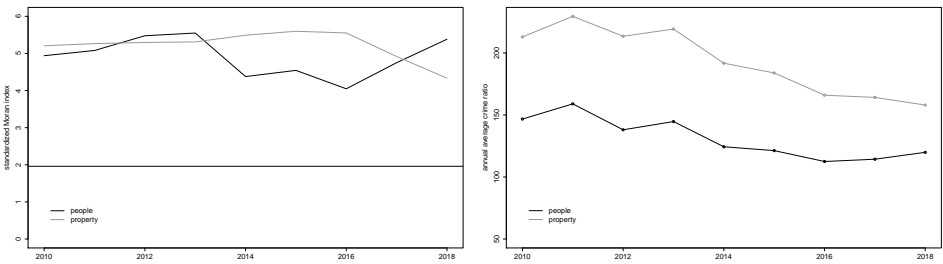

**Figure 2.** Moran indices (**left**) and annual averages (**right**) for the ratios of both crimes type (2010–2018).

Moran indices require a matrix of spatial weights to be obtained. Since the crimes are located at the commune level, this matrix has been obtained from a neighborhood structure that defines as *neighbors* those communes that share a border, following the usual criterion for irregular grids. The resulting neighborhood matrix *W* takes the form,

$$w_{ij} = \begin{cases} 0, & i = j = 1, \ldots, n; \\ 1/n_i, & \text{if } j \in V(i), \text{ with } n_i = \#V(i); \\ 0, & \text{if } j \notin V(i), \end{cases} \tag{1}$$

where $i$ and $j$ represent two communes and $V(i)$ the set of neighbours of $i$. With this structure no commune is a neighbour of itself. Note that the values in each row sum to unity because the weights, $w_{ij}$, are standardised. For other neighbourhood structures, see Cliff and Ord (1973). The packages rgdal (Bivand et al. 2021) and spdep (Bivand and Wong 2018) for the R software allow us to obtain $W$ from a shapefile with the boundaries of the MRS communes.

## 3. Spatial Panel Models

Let us start by recalling the structure of a spatial panel of data. We have $N$ spatial units that have been observed over $T$ periods. Their neighbourhood structure is known through the $W_N$ matrix of dimension $N$, described in (1). These observations involve a dependent variable $Y$, a vector of dimension $NT \times 1$ and a set of $k$ independent variables $X$, a matrix of $NT \times k$.

Models for such a data structure aim to capture the spatial interactions between spatial units over time. Their general expression is (Millo and Piras 2012),

$$y = \lambda(I_T \otimes W_N)y + X\beta + u, \tag{2}$$

or, if we include exogenous effects of the covariates (Elhorst 2014),

$$y = \lambda(I_T \otimes W_N)y + X\beta + (I_T \otimes W_N)X\theta + u, \tag{3}$$

with $I_T$ the identity matrix $T \times T$, and where the error term $u$, in its fullest expression takes the form,

$$u = (i_t \otimes I_N)\mu + \rho(I_T \otimes W_N)\varepsilon + \nu. \tag{4}$$

Some aspects of (2) and (3) deserve to be commented on. The term $\lambda(I_T \otimes W_N)y$ captures the influence of first-order neighbours on the observed dependent variable in a spatial unit. The first summand of (4), $(i_t \otimes I_N)\mu$, with $i_t$ the unit vector $T \times 1$ and $I_N$ the identity matrix $N \times N$, captures the individual-specific effects *mu* associated with each spatial unit, which are time-invariant and spatially uncorrelated. The second one, $(I_T \otimes W_N)\varepsilon$, is a vector of autocorrelated spatial errors.

It should be noted that (3) + (4) is the complete form of the model. Reduced versions depend on the assumptions made about the behaviour of the errors, resulting in *fixed* or *random* effects models. The *fixed effects* model is the one which involves fewer assumptions about the behaviour of the residuals, it assumes that $u$ decomposes into a fixed and a random part, with $\varepsilon_i \sim N(0, \sigma_\varepsilon^2)$. In contrast, the *random effects* model assumes that the $\mu_i \sim IID(0, \sigma_\mu^2)$.

There is extensive literature on the choice of type effects. A small sample would be the working paper by Granados (2011) and the texts by Croissant and Millo (2019) and Elhorst (2014). In their book, Croissant and Millo (2019) (p. 40) point out that *"The individual effects are not fixed or random by nature"* and suggest one or other type depending on the size of the panel, fixed when it is a macro-panel with an almost exhaustive sample, and random when it is a micro-panel with a random sample of a large population. Elhorst (2014), in the chapter on spatial panel models, explains the *popularity* of random models on the basis of three reasons (p. 54 ff.), but then adds that *"Despite its popularity, the question whether the random effects model is also an appropriate specification is often left unanswered"* and ends stating that *"In conclusion, we can say that the fixed effects model is generally more appropriate than the random effects model since spatial econometricians tend to work with space-time data of adjacent spatial units located in unbroken study areas, such as all counties of a state or all regions in a country"*. More on this issue can be found in Baltagi (2005), Section 2.3.1.

Taking into account the nature of our data, crimes observed in *all* the communes of the MRS over the period 2010–2018, we will opt for a fixed-effects model for our analysis, the *fixed effects spatial lag model*, whose expression is

$$y = \lambda(I_T \otimes W_N)y + (i_t \otimes I_N)\mu + X\beta + +(I_T \otimes W_N)X\theta + \varepsilon. \tag{5}$$

If the corresponding LM test suggested the alternative spatial error model, *fixed effects spatial error model*, we would resort to it. This model takes the form,

$$y = (i_t \otimes I_N)\mu + X\beta + \rho(I_T \otimes W_N)u + \varepsilon. \tag{6}$$

The first of these models, (5), relates the value of the response variable in a spatial unit to the weighted average of the values it takes in neighbouring spatial units, $(I_T \otimes W_N)y$. The spatial error model, (6), captures in the term $(I_T \otimes W_N)u$ the influence on the response variable of explanatory variables not included in the model.

## 4. Results

Before including the covariates regarding the communes in the models (5) or (6), we have obtained the correlation matrix in order to eliminate those showing high correlations. The correlations have been obtained with the data for the whole period taken together. Figure 3 represents numerically and graphically the absolute values of this matrix. The highest correlation, between ipp and women, is $cor(ipp, women) = 0.6$ and this not being excessively high, we will include all the covariates in the model.

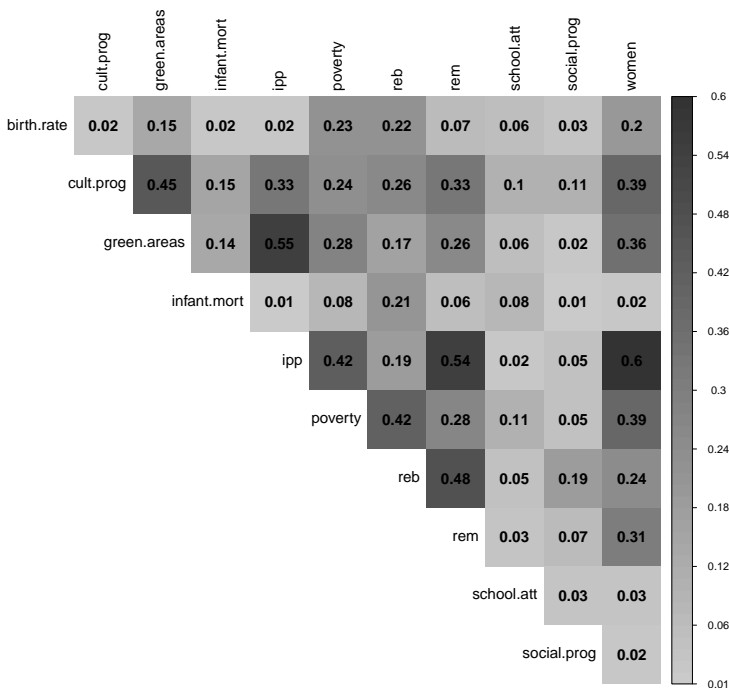

**Figure 3.** Correlation matrix associated with the variables related to the communes.

The expression of the full model fitted, considering all the available covariates, is,

```
log(ratio.per) ~ women + infant.mort + birth.rate +
        social.prog + cult.prog + ipp + school.att +
        reb + rem + poverty + green.areas + lag(women) +
        lag(infant.mort) + lag(birth.rate) + lag(social.prog) +
        lag(cult.prog) + lag(ipp) + lag(school.att) + lag(reb) +
        lag(rem) + lag(poverty + lag(green.areas) +
        log(ratio.per.1) + log(ratio.per.11) +
```

$$\text{log(ratio.prop.1)} + \text{log(ratio.prop.11)},$$

where *ratio.per* and *ratio.prop* stand for the number of crimes against people and property per 10,000 inhabitants, respectively. The suffix *.1* refers to the previous year's value, while the suffix *.11* indicates the average of the values observed in the previous year among neighbours of order 1. The inclusion of the last two variables in the model seeks to detect the broken-windows effect that may exist for crimes against property with respect to crimes against people.

Two reduced models, derived from the full model, have also been fitted. The *reduced model 1* is obtained by removing the lags of the covariates, the *reduced model 2* is obtained by removing also the spatial and spatio-temporal lags of the ratios of crimes against property.

Table 6 shows the fitting result for full model (5) and reduced modeles 1 and 2, obtained using the function *spml* from package splm (Millo and Piras 2012). In the table, the grey rows indicate the significant variables in the three models (*p*-value < 0.1), whose coefficients, particularly those associated with the lags of the ratio of crimes against people and with the influence of the environment, *lambda*, have similar values in the three models. It should also be noted that there are no major differences in the behaviour of the three models, so we will focus our attention on the most parsimonious model, the reduced model 2. As the corresponding robust LM test has a *p*-value of 0.1226, we do not need to consider the spatial error model (6). The $\lambda$ coefficient of spatial lag of the dependent variable is positive and significative, implying that whatever happens in its neighbourhood influences the commune, that is, crimes against people increase in a commune if they have happen in its neighbourhood.

**Table 6.** Estimation of spatial panel models.

| Variable | Main Model | | Reduced Model 1 | | Reduced Model 2 | |
|---|---|---|---|---|---|---|
| | $\hat{\beta}$ | *p*-Value | $\hat{\beta}$ | *p*-Value | $\hat{\beta}$ | *p*-Value |
| women | −0.0259 | 0.0568 | −0.0140 | 0.2662 | −0.0128 | 0.3095 |
| infant.mort | 0.0036 | 0.0414 | 0.0049 | 0.0062 | 0.0050 | 0.0053 |
| birth.rate | 0.0285 | 0.0000 | 0.0275 | 0.0000 | 0.0272 | 0.0000 |
| social.prog | 0.0028 | 0.2996 | 0.0028 | 0.2884 | 0.0027 | 0.3315 |
| cult.prog | 0.0089 | 0.3986 | 0.0116 | 0.2692 | 0.0114 | 0.2787 |
| ipp | −0.0085 | 0.9028 | −0.0367 | 0.5938 | −0.0635 | 0.3221 |
| school.att | −0.0012 | 0.1905 | −0.0019 | 0.0122 | −0.0020 | 0.0087 |
| reb | −0.0038 | 0.8956 | −0.0107 | 0.7147 | −0.0089 | 0.7590 |
| rem | 0.0073 | 0.2251 | 0.0092 | 0.1236 | 0.0105 | 0.0740 |
| poverty | 0.0041 | 0.0099 | 0.0028 | 0.0660 | 0.0025 | 0.0946 |
| green.areas | 0.0109 | 0.0164 | 0.0085 | 0.0670 | 0.0093 | 0.0358 |
| lag(women) | 0.0769 | 0.0200 | | | | |
| lag(infant.mort) | 0.0045 | 0.3254 | | | | |
| lag(birth.rate) | −0.0001 | 0.9926 | | | | |
| lag(social.prog) | 0.0106 | 0.1076 | | | | |
| lag(cult.prog) | −0.0316 | 0.2741 | | | | |
| lag(ipp) | 0.1982 | 0.2365 | | | | |
| lag(school.att) | −0.0021 | 0.2645 | | | | |
| lag(reb) | −0.0074 | 0.9089 | | | | |
| lag(rem) | 0.0191 | 0.2083 | | | | |
| lag(poverty) | −0.0083 | 0.0072 | | | | |
| lag(green.areas) | 0.0106 | 0.3626 | | | | |
| log(ratio.per.1) | 0.3516 | 0.0000 | 0.3775 | 0.0000 | 0.3975 | 0.0000 |
| log(ratio.per.11) | −0.3298 | 0.0009 | −0.2423 | 0.0089 | −0.3222 | 0.0056 |
| log(ratio.prop.1) | 0.0575 | 0.1954 | 0.0354 | 0.4144 | | |
| log(ratio.prop.11) | −0.0534 | 0.5546 | −0.1098 | 0.1509 | | |
| $\lambda$ | 0.4150 | 0.0000 | 0.4389 | 0.0000 | 0.4370 | 0.0000 |
| Robust LM test | 1.2449 | 0.2645 | 3.4165 | 0.0645 | 2.3841 | 0.1226 |
| $R^2$ | | 0.9560 | | 0.9538 | | 0.9535 |

Among the covariates, *infant.mort*, *birth.rate*, *school.att*, *poverty* and *green.areas* have a significant influence on crimes against people, all of these excluding *school.att* with positive coefficients. With respect to the effect that the incidences of both types of crime have on the dependent variable, crimes against people in the previous year, *log(ratio.per.1)*, have a significative and positive effect. Additionally, significative but negative is the influence of this type of crime in the previous year in the neighbourhood. With respect to covariates linked to the incidence of crimes against property, they where removed from the reduced model 2 because they show no influence, which means that the phenomenon *broken-windows* is not present.

Table 7 shows the Moran indices for the residuals of the model reduced 2. It is found that the spatial autocorrelation, which the initial data showed, disappears in the residuals.

**Table 7.** Standardised values of Moran's index for spatial panel model residuals.

| Year | Moran I | *p*-Value |
|------|---------|-----------|
| 2011 | −0.0070 | 0.5028 |
| 2012 | 0.9825 | 0.1629 |
| 2013 | −0.1089 | 0.5434 |
| 2014 | 0.3450 | 0.3650 |
| 2015 | −0.3009 | 0.6182 |
| 2016 | −0.6386 | 0.7384 |
| 2017 | −0.6672 | 0.7477 |
| 2018 | −0.5003 | 0.6916 |

## 5. Discussion

The spatial panel model with fixed-effects (5), also known as the spatial Durbin panel model, adequately describes the crime rates associated with the crimes of greatest social alarm, which we have denoted as crimes against people. The reduced version, in which the lags of the covariates were eliminated along with the variables relating to crime against property, fulfils this function equally well. Both types of crimes are part of the database corresponding to the years 2010–2018, covering a large number of unlawful conducts, mainly affecting life, the physical or psychological integrity of people, personal freedom or individual security, and property (Fiscalía de Chile 2021) (Table 4). The geographical scope of these data covers the MRS.

Among the covariates included in the model, *infant.mort*, *birth.rate*, *poverty* and *green.areas* have a significant positive influence on the rate of crimes against people, while *school.att* has a negative impact. The influence of the first three was to be expected as proposed by Núñez et al. (2003), who state that the participation of individuals in illegal activities corresponds to scarcely rational behavior, which is the product of a predisposition of individuals towards crime, which may be due to the subject's characteristics, as well as to factors in their family and social environment. In contrast to the theory formulated by Becker (1968) and Ehrlich (1973), known as "the economics of crime", which explores criminality based on economic-rational incentives. The latter in agreement with Cornish and Clarke (1986), who put forward the well-known "rational choice" theory.

As we pointed out in the Introduction, there is also abundant literature on the relationship between poverty and crime, of which these three references are a good example: Sharkey et al. (2016), Anser et al. (2020) and Dong et al. (2020). Infant mortality can be considered as a "proxy" of poverty and through this an influence on criminality (Messner et al. 2010). On the other hand, the influence of birth rate is explained, according to some authors, to the extent that the improvement in individuals' economic conditions usually leads to a lower birth rate (Sinding 2009). The influence of the variable that reflects the effect of green areas is not so evident, in fact, literature with contradictory results can be found. Thus, Shepley et al. (2019) conclude through a meta-analysis that access to nature has a mitigating effect on violence in urban settings, but Groff and McCord (2012) find that parks are associated with higher levels of crime in the surrounding area, although some specific characteristics of parks are associated with lower levels of crime. Although

not directly related to the influence of this variable on crime, several studies, including Marquet et al. (2020) are concerned with the association between objective measures of types and location of crime and public park use behaviours. The crime-reducing effect of education, measured here by the percentage of school attendance, is well known and is consistent with numerous studies on the subject (Gleditsch et al. 2022; Hjalmarsson and Lochner 2012; Machin et al. 2011).

In summary, and as Bronfenbrenner (1981) states in their ecological theory, the environment in which an individual develops and moves is composed of a set of interrelated systems that are organised at different levels, according to the proximity to the subject; where each level successively contains the other (macrosystem, exosystem, mesosystem and microsystem). It is clear that it is in this environment where the risk and protective factors that influence criminal behaviour coexist, with greater intensity in the systems closest to the indivudual. The results obtained in our research contribute to the knowledge of the different habitat factors that interact in the propensity or inhibition of the typical behaviours under analysis.

The spatio-temporal covariates derived from the dependent variable itself, *log(ratio.per)*, are influential, either what happened the previous year in the commune itself, *log(ratio.per.1)*, or in neighbouring communes, *log(ratio.per.11)*. Both appear with similar values but with different signs, positive for the former and negative for the latter, which could be interpreted as a shift of crime to neighbouring areas, which would diminish its presence in the commune. On the other hand, since crimes against property do not exert any influence, we should rule out the presence of the *broken-windows* phenomenon and we cannot affirm that this kind of crime that occurred the previous year in the same commune or neighbouring communes, foretell an increase in crimes against people. This result is in contradiction with those obtained by Glaser et al. (2021) and Liesenfeld et al. (2017) when analysing crime in Pittsburgh at census tract level in the period January-2008 to December-2013. This contradiction may only be apparent, insofar as the crimes they refer to as Part II crimes include much less serious crimes than those we refer to as crimes *against property*.

A comment on the spatial lag model. As Anselin (2005) points out, this model is suitable for describing situations in which events in the environment influence the value of the observed variable in a given geographical area, an interaction that can be assumed in crime problems.

The results obtained represent a new contribution to the study of the broken-windows phenomenon, having considered a greater number of determinants than in some previous studies, and statistical techniques of an equal or higher level of sophistication. Regarding the generalizability of our results to other contexts, we must be cautious. Previous studies on quantitative criminology show that risk factors for crime can vary considerably between cities (Connealy 2020). In any case, similar results would only be expected in regions of comparable size and characteristics.

From all of the above, it is necessary to emphasise that the conclusions expressed in this article are criminological characterisations that should be taken as one more input when creating crime prevention strategies. The success and impact of these strategies will be determined by the collaborative, inter-institutional and interdisciplinary approach that converge synergistically to attack complex criminal behaviour and its causes.

**Author Contributions:** Conceptualization, P.C.-U., Á.B.-R. and F.M.; Data curation, P.C.-U.; Methodology, F.M.; Software, Á.B.-R.; Writing original draft, F.M.; Writing review & editing, P.C.-U. and Á.B.-R. All authors have read and agreed to the published version of the manuscript.

**Funding:** This research received no external funding

**Data Availability Statement:** The data used in this analysis can be provided on request to the authors.

**Conflicts of Interest:** The authors declare no conflict of interest.

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
