# Peer review of "Crime Analysis of the Metropolitan Region of Santiago de Chile: A Spatial Panel Data Approach"

_socsci, doi:10.3390/socsci11100443_

Round 1

Reviewer 1 Report

Excellent paper. I enjoyed reading it. Some useful conclusions.

1. In section 2 (Data) might be helpful to other researchers to acknowledge  there are well-recognised potential sources of error in crime data recording (e.g. where a single crime has multiple aspects and Police KPIs of that time  can influence which heading the crime is located under).

2. Concerning the discussions around Figure 2, it might be helpful to note that in line with the last 100 years in jurisdictions around the world there is a significant trend downwards in crime rates per 10,000 population.

Author Response

Changes and corrections in the new version of the article are highlighted in red.

  1. In section 2 (Data) might be helpful to other researchers to acknowledge  there are well-recognised potential sources of error in crime data recording (e.g. where a single crime has multiple aspects and Police KPIs of that time  can influence which heading the crime is located under).

A paragraph commenting on this fact has been added before Table 4 (lines 128-142)

  1. Concerning the discussions around Figure 2, it might be helpful to note that in line with the last 100 years in jurisdictions around the world there is a significant trend downwards in crime rates per 10,000 population.

This comment has been added after Figure 2 (lines 159-161).

Reviewer 2 Report

The paper explores empirically the determinants of crimes against people across municipalities over the period 2010-2018 using a spatial panel data methodology for the Metropolitan Region of Santiago de Chile. Through their estimation, the authors aim to shed new light on the determinants of crimes against people, and investigate the broken windows theory in this context. Overall, the paper deals with an important topic. However, as explained below, there are several issues with the paper in its current state.

-Motivation. Crime is highlighted as a serious issue for several reasons, including because accelerated urbanisation has encouraged rising crime, which is supported to an extent by evidence about the public perception of crime. Whilst this is an important issue, the hypothesized rise in crime should be evidenced more clearly with reference to facts. For example, crime actually appears to be falling in the sample (Table 5). Also, is there anything else about this region/country, which makes it a useful case for analysis?

-Relative contributions. In the Introduction there is mention of differences of this study from antecedents in the literature, e.g. previous studies include fewer determinants and different statistical techniques. The relative contributions should be stated more clearly and explicitly in the Introduction. The authors should also highlight why these contributions are valuable (e.g. what is the value of adding more determinants to the model or applying a different statistical technique?).

-Estimation approach. Various estimations techniques are mentioned when discussing the literature and there is some useful discussion supporting the paper’s empirical strategy, but it should be more critical about model choice/estimation strategy over alternative approaches. Given that the central econometric objective of the paper is to identify the drivers of crime, how fit-for-purpose is the spatial panel data model used here? Does the model help overcome endogeneity issues, such that causality can be inferred from the estimation?

-Measurement errors. The sample may seem sufficiently large from a statistical perspective, but the authors should bear in mind that a sizeable proportion (the majority?) of crime goes unreported. This introduces errors into the regression model, and might explain why many of the drivers are found to be insignificantly related to crime in what is a good-sized sample. Another issue is that the reporting of crime might change over time, differ across areas, and differ by type of crime. Have these issues been considered?

-Sample. Is the panel perfectly balanced? If not, what proportion of data is missing? How have any missing data been treated, if that is the case?

-Empirical results and robustness. Only one estimation is conducted in Table 6 and the authors could consider some variations in model specification for comparison and robustness, since the contribution of this paper is essentially empirical. The authors could, for example, add other columns in the table demonstrating how the main model improves over simpler specifications which exclude the spatial and/or temporal lags (if that is the case); whether inclusion of fixed effects is worthwhile over a pooled alternative, what they add to model fit, whether they are helpful in alleviating omitted variables bias, for instance. More generally, how sensitive are the results to variation in the control set? This might help demonstrate robustness further. Also, are the results sensitive to the chosen measure of crime and its components?

-Significance of results. Can the authors be more precise about the interpretation and economic significance of the point estimates; or how would one make meaningful inferences in principle for this kind of model? Some estimates are significant statistically, but the reporting could be more precise with respect to level of significance. What is the external validity of the results? Are they applicable to other countries/regions?

Author Response

Answer Reviewer#2

Changes and corrections in the new version of the article are highlighted in red, except for new Table 6.

The paper explores empirically the determinants of crimes against people across municipalities over the period 2010-2018 using a spatial panel data methodology for the Metropolitan Region of Santiago de Chile. Through their estimation, the authors aim to shed new light on the determinants of crimes against people, and investigate the broken windows theory in this context. Overall, the paper deals with an important topic. However, as explained below, there are several issues with the paper in its current state.

-Motivation. Crime is highlighted as a serious issue for several reasons, including because accelerated urbanisation has encouraged rising crime, which is supported to an extent by evidence about the public perception of crime. Whilst this is an important issue, the hypothesized rise in crime should be evidenced more clearly with reference to facts. For example, crime actually appears to be falling in the sample (Table 5). Also, is there anything else about this region/country, which makes it a useful case for analysis?

A new paragraph, lines 29-39, responds to this question.

-Relative contributions. In the Introduction there is mention of differences of this study from antecedents in the literature, e.g. previous studies include fewer determinants and different statistical techniques. The relative contributions should be stated more clearly and explicitly in the Introduction. The authors should also highlight why these contributions are valuable (e.g. what is the value of adding more determinants to the model or applying a different statistical technique?).

See paragraph lines 70-78.

-Estimation approach. Various estimations techniques are mentioned when discussing the literature and there is some useful discussion supporting the paper’s empirical strategy, but it should be more critical about model choice/estimation strategy over alternative approaches. Given that the central econometric objective of the paper is to identify the drivers of crime, how fit-for-purpose is the spatial panel data model used here? Does the model help overcome endogeneity issues, such that causality can be inferred from the estimation?

The existence of endogeneity issues in our analysis (omitted variable, reverse causality and measurement error) cannot be totally discarded, as usual. In any case, the use of random effects in the model can alleviate potential problems regarding the presence of omitted variables and measurement errors. Besides, we should note that the main purpose of this type of econometric model is to allow us to identify certain crime risk factors or, in other words, associations between the level of crime and the covariates of interest. To infer causal relationships, it would be necessary to conduct other types of analysis under the "experimentalist paradigm" (Gibbons and Overman, 2012).

Gibbons, S., & Overman, H. G. (2012). Mostly pointless spatial econometrics? Journal of Regional Science, 52(2), 172-191.

-Measurement errors. The sample may seem sufficiently large from a statistical perspective, but the authors should bear in mind that a sizeable proportion (the majority?) of crime goes unreported. This introduces errors into the regression model, and might explain why many of the drivers are found to be insignificantly related to crime in what is a good-sized sample. Another issue is that the reporting of crime might change over time, differ across areas, and differ by type of crime. Have these issues been considered?

The problem of unreported crimes and the error they introduce into the regression models is discussed in a new paragraph added before Table 4 (lines 128-142).

-Sample. Is the panel perfectly balanced? If not, what proportion of data is missing? How have any missing data been treated, if that is the case?

The panel is perfectly balanced and we have explicitly stated this in the Section Data (line 127)

-Empirical results and robustness. Only one estimation is conducted in Table 6 and the authors could consider some variations in model specification for comparison and robustness, since the contribution of this paper is essentially empirical. The authors could, for example, add other columns in the table demonstrating how the main model improves over simpler specifications which exclude the spatial and/or temporal lags (if that is the case); whether inclusion of fixed effects is worthwhile over a pooled alternative, what they add to model fit, whether they are helpful in alleviating omitted variables bias, for instance. More generally, how sensitive are the results to variation in the control set? This might help demonstrate robustness further. Also, are the results sensitive to the chosen measure of crime and its components?

Table 6 now includes two new reduced models derived from the full model as suggested by the reviewer. This implies changes in the presentation of the results and in the Conclusions (lines 234-243, 263-265, 294-297).

-Significance of results. Can the authors be more precise about the interpretation and economic significance of the point estimates; or how would one make meaningful inferences in principle for this kind of model? Some estimates are significant statistically, but the reporting could be more precise with respect to level of significance. What is the external validity of the results? Are they applicable to other countries/regions?

See the new paragraph at the end of the conclusions (lines 324-330).

Round 2

Reviewer 2 Report

I am happy with the revisions made which address the main points raised in my review.